# Hybrid Sausages Using Pork and Cricket Flour: Texture and Oxidative Storage Stability

**DOI:** 10.3390/foods12061262

**Published:** 2023-03-16

**Authors:** Xiaocui Han, Binbin Li, Eero Puolanne, Marina Heinonen

**Affiliations:** Department of Food and Nutrition, University of Helsinki, Agnes Sjöbergin katu 2, 00790 Helsinki, Finland

**Keywords:** cricket flour, hybrid sausage, texture, lipid oxidation, protein oxidation

## Abstract

This study aimed to study the functionalities of cricket flour (CF) and the effects of the addition of CF on the texture and oxidative stability of hybrid sausages made from lean pork and CF. Functional properties of CF, including protein solubility, water-holding capacity, and gelling capacity, were examined at different pHs, NaCl concentrations, and CF contents in laboratory tests. The protein solubility of CF was significantly affected by pH, being at its lowest at pH 5 (within the range 2–10), and the highest protein solubility toward NaCl concentrations was found at 1.0 M (at pH 6.8). A gel was formed when the CF content was ≥10%. A control sausage was made from lean pork, pork fat, salt, phosphate, and ice water. Three different hybrid sausages were formulated by adding CF at 1%, 2.5%, and 5.0% levels on top of the base (control) recipe. In comparison to control sausage, the textural properties of the CF sausages in terms of hardness, springiness, cohesiveness, chewiness, resilience, and fracturability decreased significantly, which corresponded to the rheological results of the raw sausage batter when heated at a higher temperature range (~45–80 °C). The addition of CF to the base recipe accelerated both lipid and protein oxidation during 14 days of storage, as indicated by the changes in TBARS and carbonyls and the loss of free thiols and tryptophan fluorescence intensity. These results suggest that the addition of CF, even at low levels (≤5%), had negative effects on the texture and oxidative stability of the hybrid sausages.

## 1. Introduction

The latest estimation by the Food and Agriculture Organization of the United Nations (FAO) shows continuously growth of the protein demand around the world due to population growth [1]. Lean meat is considered protein-rich, with an average protein content of about 22%, and has a balanced essential amino acid composition, playing an essential role in the human diet [2]. However, the increasing global production and consumption of meat has triggered global environmental concerns in terms of greenhouse gas emissions, land use, pollution, and biodiversity loss [3,4,5,6]. To address this issue, calls for a reduction in meat consumption and an increase in the production of alternative proteins have been proposed [7].

In this respect, insects have gained interest as sustainable protein sources, mainly due to their environmental and nutritional benefits. Insect farming, compared to farming cattle or pigs, generally requires less feed, water, and land, and emits fewer greenhouse gases and less ammonia [8]. Insect rearing for food has a high feed-to-protein conversion efficiency: for example, house crickets (*Acheta domesticus*) are at least 4 times more efficient than pigs and 12 times more efficient than cattle [8,9]. From a nutritional perspective, *A. domesticus* has a crude protein content of ~13–25% as a wet basis and ~42–75% as dry matter [10]. All essential amino acids are present in *A. domesticus,* and the contents of essential amino acids are comparable to those of pork and beef [11,12]. Additionally, *A. domesticus* has a protein digestibility-corrected amino acid score of 0.69–0.92, which is comparable to the 0.92 score of meat and higher than those of plant proteins such as lentils, beans, peas, and chickpeas, which have scores of 0.57–0.71 [13]. The digestible indispensable amino acid score of *A. domesticus* was found to be >75, with sulfur amino acids (cysteine and methionine) identified as the first limiting amino acids for children, adolescents, and adults [14]. 

There are 2111 edible insect species consumed by 2 billion people in 113 countries, most species being collected wild, and a few species being farmed [15,16]. Entomophagy (the practice of eating insects) is already a part of the diets in some regions, and insects are even considered delicacies in many cultures located in Africa, Asia, and South America [8]. In Europe, however, consumer acceptance of eating insects is still a challenge, due to reasons such as cultural taboos and safety concerns (possible toxicity, allergenicity, or microbial loads) [17,18]. Research suggests that adding insects, in such a way that they become invisible in food preparations, might help to reduce insect food neophobia [19]. For example, insects could be incorporated into hybrid meat products, which may be used to persuade predominant meat eaters to reduce their meat consumption. 

Past studies incorporating insect-based ingredients into processed meat products have focused on the compositional and technological aspects of hybrid products. Insect flour from mealworms (*Tenebrio molitor*), silkworms (*Bombyx mori* pupae), and house crickets (*A. domesticus*) have been reported to be suitable to replace up to 20% of the pork in cooked sausages or patties without negative effects on nutritional or technological properties [20,21,22,23]. Furthermore, Kim et al. (2020) [24] suggested the use of protein extract from yellow mealworms (*T. molitor*) to replace about 40% of the pork myofibrillar protein in a hybrid emulsion system. Meat-insect hybrid products containing pork and superworms (*Zophobas morio* larvae) have been reported to have similar viscoelastic properties, water-holding capacities, and gel-forming abilities as compared to meat products, provided that a higher heating temperature and the proper insect:meat ratio were used [25,26]. However, the desired texture tends to pose a challenge when developing meat-insect hybrid products. Kim et al. (2020) and Scholliers, Steen, and Fraeye (2020) pointed out that the interaction between insect protein and myofibrillar protein resulted in degraded rheological properties and thermal stability of the hybrid emulsion system in comparison to pure meat systems [24,27]. F. Zhang et al. (2022) found a lowered sensory score for cooked sausages when replacing lean meat with defatted mealworm (*T. molitor*) flour even at a low ratio, indicating that the processing condition (heating temperature, ionic strength) is crucial for the desired texture [28]. The use of functional ingredients (such as hydrocolloids, starch, and plant-based proteins) was suggested to improve the texture of the final products when developing meat-insect hybrid products [26]. In addition to meat-insect hybrid products, combinations of plant-based materials and insects to produce hybrid products (such as extrudates, jerky, breads, pasta, 3D-printed snacks, and chocolate chip cookies) have also been suggested, aiming to improve the textural and nutritional quality of the final products under controlled processing conditions [29,30,31,32,33,34,35,36,37,38,39].

Oxidation reactions are crucial to the quality of the final products because they can result in various compounds altering the sensory, structural, and nutritional properties of foods. Currently, there are limited research papers studying the oxidative stability of hybrid products made from combined meat and insect products during storage. Kim et al. (2016) reported a slight increase in lipid oxidation during 7 days of storage of pork-insect sausages which were made by adding defatted mealworm (*T. molitor*) or silkworm (*B. mori* pupae) flour to a pork sausage recipe [22]. However, there is no existing literature regarding protein oxidation of a hybrid product combining meat and insects. Therefore, the aim of the current study was to investigate the effects of added cricket flour on the textural properties and oxidative stability of hybrid sausages in order to clarify the technological properties of insect meal as an ingredient in meat products. The specific aims were to (1) examine the functional properties of cricket flour (CF), (2) evaluate the effects of CF addition on the textural properties of cooked sausages, and (3) evaluate the effects of CF addition on lipid and protein oxidation of the cooked sausages during storage.

## 2. Materials and Methods

### 2.1. Materials

House crickets (*A. domesticus*) were purchased as whole and frozen at the adult stage from Villa Mangs Ltd. (Raasepori, Finland). After lyophilization (–80 °C, 0 mbar, ≥4 days) (Gamma 2–20 Freeze Dryer, Martin Christ, Osterode, Lower Saxony, Germany) and grinding (GM200, Retsch, Haan, North Rhine-Westphalia, Germany), CF was stored in desiccator at –20 °C in the dark until use. Freeze-drying was chosen to improve the mixing and comminution of the crickets, in order to avoid visible brown particles on the cut surface and overall browning of the insect ingredients, which would be caused by heating. Lean pork and pork fat were ordered from Lihatukku Heikin Liha Ltd. (Helsinki, Finland); after pre-chopping, lean pork and pork fat were vacuum-packed and kept at –20 °C in the dark until use. All chemicals and reagents were of analytical grade and obtained from either Sigma-Aldrich (Seelze, Germany) or Merck (Darmstadt, Germany).

### 2.2. Functional Properties of CF

The protein solubility of CF at different pH levels (2–10) and NaCl concentrations (0–2.5 M) was determined using the method outlined by Kim et al. (2017) [23]. To investigate how pH affects protein solubility, 1 g of CF was weighed in a centrifuge tube and mixed (vortex 30 s) with 10 mL Milli-Q water. The pH of the mixture was adjusted with a few drops of 6 M HCl or 6 M NaOH solution. To study the effect of NaCl concentration on protein solubility, 1 g of CF was mixed with 10 mL NaCl solutions (0, 0.25, 0.5, 1, 1.5, 2, 2.5 M) at pH 6.8. The mixture was mixed well (vortexed for 2 min) and shaken overnight (4 °C). After centrifuging (3220× *g*, 4 °C, 15 min), the protein content in the supernatant was determined according to DC Protein Assay (Bio-Rad, Hercules, CA, USA) against BSA at a wavelength of 750 nm using a spectrophotometer (UV-1800, Shimadzu, Tokyo, Japan). The protein solubility of CF was expressed as g protein in the supernatant per g sample.

Water-holding capacity of CF was determined by a laboratory method. First, 1 g of CF was mixed with Milli-Q water or NaCl solutions (0.5, 1.0, 1.5, 2.0, or 2.5 M) at pH 6.8 (vortex, 2 min, room temperature (r.t.)). The mixture was heated at 70 °C (water bath, 30 min). After cooling and centrifuging (5000× *g*, 21 °C, 30 min), the supernatant was discarded. The water-holding capacity was calculated as the weight difference between the pellet and the 1 g sample.

The gel-forming ability of CF was evaluated at 1%, 2.5%, 5%, 10%, 20%, and 30% (g/100 mL) in Milli-Q water, 1.0 M NaCl, or 2.1 M NaCl using the method by Yi et al. (2013) [40]. The mixture in the test tube was vortexed (2 min) and adjusted to pH 6.8 by slowly adding 6 M HCl or 6 M NaOH solutions. After dissolving at 4 °C overnight, gel formation was determined through visual observation after heating at 70 °C (water bath, 30 min). If the liquid mixture was not moving upon turning the test tube, it was considered as a gel.

### 2.3. Proximate Composition

Lean pork and CF were characterized by determining their moisture, protein, crude fat, and chitin contents, in triplicate. The moisture content was measured by an oven air-drying method [41]. The protein content was analyzed by the Kjeldahl method with a nitrogen-to-protein conversion factor of 5.09 for CF [42] and 6.25 for lean pork, respectively. The crude fat content was analyzed based on Soxhlet extraction using the Soxtec system (Soxtec 2050, Foss Tecator, Hillerød, Denmark). The fatty acid composition of CF was determined by the laboratory method using GC-FID (Agilent 6890 N, Agilent, Santa Clara, CA, USA). The chitin content of CF was measured according to a reversed-phase ultra-high performance liquid chromatography method outlined by Han and Heinonen (2021) using UHPLC (Acquity Ultra Performance LC, Waters, Milford, MA, USA) equipped with a fluorescence detector [43]. The proximate compositions of the sausages were calculated from the analytical values of lean pork and CF.

### 2.4. Sausage Preparation

A preliminary laboratory sausage test was carried out to find the suitable levels of meat, fat, and water for reasonable sausage quality within the variation necessitated by CF additions. The sausages of a pilot plant scale were formulated according to Table 1. The control sausage was made of 42.5% ice water, 35% lean pork, 20% pork fat, 2% salt, and 0.5% phosphate, by weight. For the hybrid sausages, CF was added at 1% (CF-1.0%), 2.5% (CF-2.5%), and 5.0% (CF-5.0%) levels on top of the base (control) recipe. The day before the preparation, lean pork, pork fat, and CF were thawed overnight at 5 °C. After thawing, lean pork was pre-homogenized with a table cutter (Seydelmann K20, Seydelmann KG, Aalen, Germany) following mixing with salt and a portion of the ice (<0 °C). Chopping was continued by adding pork fat and part of the cold tap water/ice to obtain a fine batter (6–8 °C). Subsequently, CF and the remaining cold tap water/ice were added. Further chopping of the final batter was continued until the batter temperature reached 14 °C. The uncooked batter was stuffed (VINS/22 sausage filler, la Minerva di Chiodini Mario, Bologna, Italy) into a Walsrode K flex polymer casing (45 mm in diameter), (Wolff Walsrode AG, Bomltiz, Germany) with string-closed ends. Sausages were cooked at 75 °C using liquid steam equipment (Kerres CS 350, Kerres GmbH, Backnang, Germany) for 45 min (until the sausage core temperature reached 72 °C). Finally, the cooked sausages were quickly cooled down in an ice bath and stored at 4 °C for further analyses. The preparation of the sausages of each group was conducted once to make the four different batters, and thereafter, the sampling of the sausages was repeated three times. To determine the textural properties and oxidative stability, samples were collected on days 0, 1, 4, 7, and 14 to determine the textural properties and oxidative stability during storage.

The control sausages were formulated with lean pork, pork fat, salt, phosphate, and ice water; the CF sausages were hybrid sausages formulated by adding CF at 1% (CF-1.0%), 2.5% (CF-2.5%), and 5.0% (CF-5.0%) levels to the base (control) recipe.

### 2.5. Rheological Properties of Raw Sausage Batter

The rheological properties of raw sausage batter (before cooking) were measured according to Chen et al. (2020) [44], with modifications, using a rheometer (HAAKE MARS 40, ThermoScientific, Bremen, Germany) with parallel-plate geometry (35 mm in diameter and 1 mm in gap) in oscillatory mode. A temperature ramp sweep (20–80 °C, 2 °C/min as the heating rate) was conducted. The oscillation frequency was set at 1.0 Hz. The strain (γ_0_ = 0.3%) was set according to the amplitude sweep at 20 °C for both the raw and cooked batter (after heating at 80 °C) to ensure that the measurement was conducted within the linear viscoelastic range of each sample. To prevent the sample from drying during the measurement, a sample hood with a solvent trap was used. The storage modulus (G′), loss modulus (G″) and phase angle (δ) were recorded to demonstrate the rheological properties of raw sausage batter.

### 2.6. Color and Textural Properties of Cooked Sausages

The color of the sausages was measured using a chromometer (CR-400, Konica Minolta, Osaka, Japan). Each sample was cut into slices of 1 cm thick from the middle of the cooked sausage. The color space parameters *L** (lightness), *a** (red coordinate), and *b** (yellow coordinate) were recorded immediately after the cutting of the samples. The total color difference (∆*E**) between the control and CF sausages was calculated according to the following equation:$$∆E^{*}=\sqrt{∆{L^{*}}^{2}+∆{a^{*}}^{2}+∆{b^{*}}^{2}}$$
where ∆*L** is the difference in lightness and darkness (+ = lighter, − = darker), ∆*a** is the difference in red and green (+ = redder, − = greener), and ∆*b** is the difference in yellow and blue (+ = yellower, − = bluer).

Texture profile analysis was performed using a Texture Analyzer (TA XT2 i, Stable Micro Systems, Brookfield, UK) equipped with a 30 kg load cell and a cylinder probe (36 mm in diameter). A two-cycle compression test (each sample was compressed to 40% of the original height in two consecutive cycles) was applied to each sausage sample in a regular shape (26 × 26 × 26 mm). The testing conditions were as follows: pre-test speed 1 mm/s, test speed 1 mm/s, post-test speed 5 mm/s, resting time 5 s, and trigger force 5 g. Each measurement generated a two-cycle force–time curve from which the hardness, springiness, cohesiveness, chewiness, and resilience were calculated by the program.

Fracturability was measured using an Instron device (33 R 4465, Instron, High Wycombe, UK) equipped with a 100 N load cell and a pistol probe (18 mm in diameter, 2 mm in thickness) in penetration mode. Each sausage sample (50 mm in height, 45 mm in diameter) was penetrated 30 mm (60% of the sample height) at rate 100 mm/min. Each penetration generated a force–distance curve. The maximal force (Fmax) during the penetration was recorded to represent the fracturability of the sausage.

### 2.7. Lipid Oxidation of Cooked Sausages

Lipid oxidation was monitored by measuring the thiobarbituric acid-reactive substances (TBARS) [45,46]. For this process, 5 g of sausage was mixed with 15 mL trichloroacetic acid (5%, g/100 mL) and 0.5 mL butylated hydroxytoluene (4.2% in ethanol, g/100 mL) using a homogenizer (Ultra-Turrax^®^ T25, IKA, Staufen, Germany) at 13,500 rpm for 30 s in an ice bath. The mixture was filtered (WhatmanTM 42, GE Healthcare Life Sciences, Gillingham, UK), mixed with 0.02 M thiobarbituric acid (1/1, *v/v*), and boiled in a water bath (100 °C, 40 min). After cooling, the absorbance was recorded at a wavelength of 532 nm using a spectrophotometer (UV-1800, Shimadzu, Tokyo, Japan). A standard curve of 1,1,3,3-tetraethoxypropane was used to calculate the amount of malondialdehyde produced. The TBARS contents of the sausages were expressed as mg malondialdehyde/kg sausage.

### 2.8. Protein Oxidation of Cooked Sausages

The carbonyl content was measured by the 2,4-dinitrophenylhydrazine (DNPH) assay [47,48] with slight modifications. A sausage sample of 1 g (in duplicate) was mixed well with 10 mL 0.15 M KCl using a homogenizer (Ultra-Turrax^®^ T25, IKA, Staufen, Germany) at 9500 rpm for 30 s in an ice bath. An aliquot of 100 µL of the homogenate (5 replicates) was mixed with 1 mL of 10% (g/100 mL) TCA and centrifuged (5000× *g*, 5 min, r.t.). The pellet was collected, mixed with 400 µL of 5% (g/100 mL) SDS, then subsequently heated (water bath at 100 °C, 10 min) and ultra-sonicated (water bath at 40 °C, 30 min). The mixture was treated with 0.8 mL of 0.3% (g/100 mL) DNPH in 3 M HCl, while the blank (in duplicate) sample was prepared by adding 0.8 mL of 3 M HCl. After incubation (30 min, r.t.), the proteins were precipitated by adding 400 µL of 40% (g/100 mL) TCA following centrifugation (5000× *g*, 5 min, r.t.). The pellet was collected and washed three times with 1 mL of ethanol-ethyl acetate (1:1, *v:v*) in another centrifugation (10,000× *g*, 5 min, r.t.). The resulting pellets were nitrogen-dried, then subsequently dissolved in 1.5 mL of 6 M guanidine hydrochloride in 20 mM NaH_2_ PO_4_ (pH 6.5) and incubated overnight (4 °C, in the dark). The absorbance was read at 280 and 370 nm. The carbonyl content, expressed as nmol/mg of protein, was calculated as follows:$$carbonylcontent(nmol/mgofprotein)=\frac{{Abs}_{370}-{Abs}_{370}\left(blank\right)}{22,000\left[{Abs}_{280}-\left({Abs}_{370}-{Abs}_{370}\left(blank\right)\right)\times 0.43\right]}\times 10^{6}$$
where 22,000 is the molar extinction coefficient and 0.43 is the coefficient for removing potential hydrazine interference at 280 nm.

Free thiol content was measured using the 5,5′-dithio-2-nitrobenzoate (DTNB) method [49,50]. A sample of 1 g of sausage (in duplicate) was mixed with 25 mL 5% SDS in 0.1 M Tris-HCl (pH 8.0) using a homogenizer (Ultra-Turrax^®^ T25, IKA, Staufen, Germany) at 12,800–13,500 rpm (30 s, r.t.). The homogenates were heated (water bath at 80 °C, 30 min), cooled to r.t., and filtered (Whatman^®^ filter paper, Grade 42: 2.5 μm). Thiols were measured by mixing the filtrate, 0.1 M Tris-HCl (pH 8.0), and 10 Mm DTNB in 0.1 M Tris-HCl (pH 8.0) (1/4/1, *v/v/v*). A protein blank (filtrate/0.1 M Tris-HCl (pH 8.0), 1/5, *v/v*) and reagent blank (0.1 M Tris-HCl (pH 8.0)/10 Mm DTNB in 0.1 M Tris-HCl (pH 8.0), 5/1, *v/v*) were prepared for each sample. The mixture was incubated (30 min, r.t., dark), and then absorbance was recorded at 405 nm. The concentration of thiols in the filtrate was calculated using a molar extinction coefficient of 13,600 M^−1^ CM^−1^ for DTNB at the given wavelength. The protein content in the filtrate was determined according to a DC Protein Assay, where a standard curve was established using BSA at 750 nm. The free thiol content was expressed as nmol thiol/mg protein.

The loss of tryptophan fluorescence intensity (FI) was determined according to Estévez et al. (2008) [51] using a spectrometer (LS55, PerkinElmer, Waltham, MA, USA). A sausage sample of 1 g (in duplicate) was mixed with 10 mL of ice-cold 0.15 M KCl using a homogenizer (Ultra-Turrax^®^ T25, IKA, Staufen, Germany) at 9500 rpm for 30 s in an ice bath. The mixture was filtered and diluted by Milli-Q to ~20 µg protein/mL. The emission spectra were recorded from 320 to 460 nm, with the excitation wavelength at 290 nm (slit = 10 nm, speed = 180 nm/min).

### 2.9. Statistical Analysis

All results are given as mean ± standard deviation (*n* = 3). To study lipid and protein oxidation of cooked sausages during storage, all data were analyzed by ANOVA under the general linear model using IBM SPSS Statistics software (Version 28.0.0.0 (190)), with storage days and CF addition (CF%) being fixed factors. Tukey’s test (*p* ≤ 0.05) was used to determine significant differences among the results. Pearson’s correlation test was performed between the indicators of lipid and protein oxidation.

## 3. Results and Discussion

### 3.1. Functional Properties of CF

The functional properties, including the protein solubility, water-holding capacity, and gel formation, of non-meat proteins play an important role in the quality characteristics of processed meat products [52]. Therefore, these properties were investigated to improve the sausage recipe. In this study, the protein solubility of CF was significantly affected by pH and NaCl concentration (Figure 1a,b). The lowest protein solubility in CF was observed at an acidity of pH 5 (Figure 1a), which, mostly likely, can be attributed to the isoelectric point of the predominant muscle protein actin (close to 40 kDa) in house crickets (*A. domesticus*) [53]. This result is consistent with previous studies which have reported the isoelectric point of insect proteins to be in the range of pH 4–6, depending on species and sex [23,53,54,55,56,57]. Solubilized protein in CF increased significantly when pH was >6 or <4. Particularly at pH 2–3, the protein solubility of CF was >3 times higher than at pH 5. The protein solubility increased with an increase in NaCl concentration from 0–1.5 M at pH 6.8 (Figure 1b), which is consistent with the results of Kim et al. (2017) [23]. The increase in protein solubility upon increased NaCl concentration was also observed with yellow mealworm, which can be attributed to the salt effectively allowing the protein molecules to form an electric double layer with sodium and chloride ions [58,59].

The water-holding capacity of CF (Figure 1c) was found to be ~2 g/g, which is in line with the results from the previous study of Zielińska et al. (2018) [57]. The water-holding capacity of CF increased significantly with increasing NaCl concentration, from 0 to 2.5 M, showing that CF had better efficiency in holding water molecules at increased NaCl concentrations. This value is lower than that of meat (3 g/g on dry matter) [60]. In general, the presence of salt increases the water-holding capacity of meat. One of the hypotheses is that the adsorbed Cl^−^ ions increase the negative charge of myosin, which can break down the myosin structure and increase the water-accessible surface area of the myofibrillar proteins [60,61].

The visual appearance of the gelation of CF was determined at six levels (1–30%, g/100 mL) in three NaCl concentrations, from 0–2.1 M, at pH 6.8 (Appendix A). When the CF content was ≥10%, all the mixtures formed a gel at pH 6.8 after heating at 70 °C for 30 min, without any differences observed when the NaCl concentration increased. Factors affecting the gel properties were pH, protein content, and thermal treatment. Kim et al. (2017) [23] did not find gel formation in the CF content at a range of 5–20% at pH 6; however, a large African cricket (*Grylliae* sp.) was able to form a gel at 10% content using a higher temperature and longer heating time [62]. Yi et al. (2013) [40] suggested that a higher protein content would be required to form a gel if the pH were neutral. With solubilized meat myosin, the gelling has a peak form maximum at pH 6.0, and the temperature maximal level is reached at 65 °C and onwards [63]. Further studies need to be conducted in order to fully understand the gelling behavior of CF upon heating (temperature and duration).

### 3.2. Proximate Compositions

The proximate compositions of the main ingredients (lean pork and CF) and the sausages are shown in Table 2. The proximate compositions of lean pork are in line with the results from previous meat-insect studies [26,64], with water (74.9 ± 0.2%) and protein (20.3 ± 0.4%) being the major components. The major components in CF were proteins (56.0 ± 0.5%) and crude fat (23.5 ± 0.5%), which is in line with the case reviewed by Ververis et al. (2022) [10]. It needs to be specified that the chemical compositions of insects are highly dependent on the rearing conditions (feed, developmental stage at the time of harvesting, ambient conditions, etc.) [65]. To keep one main variable in this study, a small quantity of CF was added on top of the basic recipe instead of replacing the lean meat. As the CF was in the form of dry powder, the other components, especially salt (2%) and phosphate (0.5%) (key factors affecting the protein solubility and water-holding capacity), were, thus, only slightly diluted, without any notable difference in their contents (Table 1), especially in the water phase. The proximate compositions of each CF sausage were notably different (based on the calculated proximate composition values) when compared to the control sausages. With increasing additions of CF, the moisture content decreased from 68.7 ± 0.1% to 65.8 ± 0.1%, while the protein content increased from 7.1 ± 0.1% to 9.4 ± 0.1%. A similar trend was found in previous studies, where the replacement of lean pork with house cricket, mealworm larvae, or silkworm pupae flour decreased the moisture but increased the protein content of sausages, which can be attributed to the higher solid content of the edible insect flours compared to pork [22,23]. Although the chitin level was low in the CF sausages (0.1–0.4%), it showed an increase with the increasing CF additions. Additionally, the addition of CF had almost no effect on the content of crude fat (~21%) and other components (~3%, mainly including 2.0% salt and 0.5% phosphate, according to the sausage recipe shown in Table 1) among all the sausage groups.

### 3.3. Rheological Properties of Raw Sausage Batter by Heating

The storage modulus (G′), loss modulus (G″), and phase angle (δ) were monitored during a temperature ramp sweep to reflect the heat-induced gel structure of proteins from uncooked sausage batter (Figure 2). For all the samples, G′ (elastic properties) was always higher than G″ (viscous properties) throughout the entire heating process, especially at temperature ~55 °C, indicating the formation of a heat-induced elastic gel structure (Figure 2a). For the control sausage, G′ increased slightly and reached a small peak at 33 °C due to the aggregation of myosin heads, indicating the initial formation of a loose gel-like structure. At 33–45 °C, G′ slightly decreased due to the denaturation of myosin tails. Then, G′ showed a sharp increase, from ~45 to 55 °C, followed by a steady slight increase until 80 °C due to the cross-linking between myosin tails, indicating the transformation from a viscous sol into an elastic gel network. This is in line with the heat-induced gelling behavior of myofibrillar proteins, which has been recently reviewed by Y. Zhang et al. (2022) [66]. For the CF sausages, the change of G′ followed a similar pattern as the control sausage throughout the entire heating process, indicating the formation of a gel structure upon heating. These results agree with the dynamic rheological pattern of myofibrillar protein from animals [28,44,67]. However, all CF sausages had lower G′ in the temperature range of ~50–80 °C as compared to the control, indicating that the addition of CF resulted in a weaker gel. With increasing CF%, hybrid sausages with similar lean meat contents (33–35%, Table 1) showed decreased elasticity, indicating a weakened heat-induced myofibrillar gelation towards increasing CF levels.

In a higher temperature range (~55–80 °C), the phase angle (δ) of each sausage group decreased to <15°, demonstrating the formation of an intact elastic structure (Figure 2b). Compared to the control, all CF sausages had higher δ when the temperature was above ~55 °C, showing slightly decreased elastic gel strength due to added CF. These results are consistent with G′ and G″, indicating that added CF has a negative effect on building a strong 3D cross-linked gel network. It is hypothesized that the polyphenol oxidase originating from CF catalyzed the oxidation, resulting in the formation of oxides that further react with sulfhydryl groups in proteins, preventing the formation of stable disulfide bonds among the proteins to form a stable gel [28,40,68]. This assumption is supported by the free thiol contents which were analyzed (Figure 3c). Interestingly, CF-5.0% showed a steady, albeit slight, decrease in δ (~17–14°) over the whole temperature range (20–80 °C), being at its highest in higher temperature ranges (~55–80 °C). This can be explained since the added CF significantly decreased the moisture of the batter (Table 2), leading to a reduction in interaction within the protein network and further lowering of the rigidity of the protein gel.

### 3.4. Color and Texture

#### 3.4.1. Color of Cooked Sausages

Color differences between the control and CF sausages were greatly affected by adding CF at different levels (Table 3). As compared to the control sausage, the total color difference was clearly visible (Δ*E** > 6.0) when CF addition was ≥2.5%. CF sausages were confirmed to be darker compared to the control sausage by visual observation (Appendix A). With increasing CF levels, the lightness (*L**) significantly decreased, while the redness (*a**) and yellowness (*b**) values increased significantly. These results agree with those obtained when mealworm was used to partially replace pork in sausages [21,24,28]. The black, brown, and yellow colors typical of insects are attributed to the cuticular protein melanin [69]. Therefore, the color of the sausages can be greatly affected by melanin levels even when CF is added at a low level (≤5%).

#### 3.4.2. Texture of Cooked Sausages

With increasing CF levels, the textural properties, namely, hardness, springiness, cohesiveness, chewiness, resilience, and fracturability, of the CF sausages clearly decreased as compared to the control sausage (Table 3). This indicated the inferior texture of the hybrid sausages as compared to that of control sausage. The visual observation of CF sausages also showed a paste-like structure, followed by an increasing CF addition (Appendix A). The cooking temperature of the sausages was 75 °C (Section 2.4 Sausage preparation). Therefore, these findings are in accordance with the rheological values obtained at the higher temperature range (~55–80 °C), where the control sausage showed a stronger gel strength than the hybrid sausages (Figure 2). This result is in accordance with previous findings regarding the addition of mealworm to meat sausages followed by reduced hardness, springiness, chewiness, resilience, and fracturability [21,24,28]. Schollier et al. (2020 a, b) [25,26] also found that the maximum force measured during penetration was reduced in hybrid sausages. The stronger structure of the control meat sausage was most likely due to the intermolecular interactions of myosin molecules, which have been recently reviewed by Y. Zhang et al. (2022) [66]. Upon heating, solubilized myosin molecules unfold, then interact with each other through exposed active binding sites such as (1) hydrophobic groups to form corresponding hydrophobic bonds; (2) thiols to form disulfide bridges, leading to the formation of protein gel network and eventually contributing to the desired sausage texture, including firmness, gumminess, and chewiness. Additionally, (3) hydrogen bonds formed between carbonyl oxygen and amino hydrogen can stabilize the α-helical structure of myosin tails within the native protein molecules; and (4) electrostatic interactions due to the polar groups also contribute to the gel network, depending on the conditions of pH, salt, and temperature. In this study, the inferior structure of CF sausages might be due to the insect cuticular components, chitin and quinone-tanned scleroproteins, which are robust and resistant to chemical and enzymatic degradation [71]. In insect procuticles, microfibrils formed by chitin molecules are embedded side-by-side in the proteins to make sheets. In epicuticles, protein chains are chemically cross-linked to each other by quinones, leading to irreversible stiffening and, thus, creating support structures for insects. These components are highly hydrophobic and they may interfere with and lower the protein gel network generated by myofibrillar proteins.

However, the opposite effect was found in hybrid sausages (using house cricket, mealworm, or silkworm flour), with increased hardness, springiness, cohesiveness, gumminess, and chewiness compared to the control meat group [22,72]. Kim et al. (2016) [22] pointed out that hardening will be inevitable if meat is partially replaced by insect flours (mealworm larvae and silkworm) because of the decreased moisture content and increased solid compounds. It must be noted that the pre-pretreatments of insect flours affect the protein functionality, which results in altered texture of the final hybrid products. Kim et al. (2017), Choi et al. (2017), and F. Zhang et al. (2022) [21,23,28] found that the structures of hybrid products improved when using spray-, vacuum-, or microwave-dried insect flour compared to freeze-dried insect flour. In general, different temperatures and pressure levels during processing cause different degrees of protein denaturation. Most likely, an extended protein denaturation process was triggered by the freeze-drying stress and the longer duration of storage, which decreased the sulfhydryl group content, as a higher sulfhydryl content is crucial to form a stronger heat-induced protein gel matrix by reinforcing the intermolecular network [28,73].

Chemical and enzymatic modifications can be used to optimize insect flour for comminuted meat products. Kim et al. (2016) [22] found that acid-hydrolyzed mealworm larvae and silkworm pupae flour had a positive effect on the texture of hybrid sausages. Incorporation of transglutaminase into the hybrid sausage also brought positive structural changes compared to the control sausage, which can be attributed to the cross-linkage (glutamine and lysine in protein molecules) between the transglutaminase and the non-meat proteins [72]. However, how these treatments affect the protein functionality of insect flour due to protein conformational changes and denaturation has not been studied so far. Additionally, the knowledge of protein profiles in various insect species is lacking, which is vital to understand the mechanisms behind the protein functionalities and the changes they undergo during processing.

### 3.5. Oxidative Stability of Cooked Sausages during Storage

#### 3.5.1. Lipid Oxidation

Both the length of the storage and the amount of added CF (CF%) had a pronounced effect (*p* < 0.001) of an increase in TBARS during storage, and a significant storage day × CF% interaction was observed (*p* < 0.001) (Figure 3a, Table 4). On Day 0, with increasing CF addition, TBARS values were remarkably higher compared to the control sausage. During storage, the TBARS content remained stable during the first 7 days, but dramatically increased from day 7 to 14 for the CF sausages. The TBARS value also gradually increased in the control sausage within 14 days, but at a much lower level compared to that of the CF sausages. These results indicate that CF is able to accelerate the accumulation of secondary lipid oxidation products in hybrid sausages during storage, which is consistent with the findings from Kim et al. (2016) [22]. The accelerated lipid oxidation, especially after one week, in CF sausages may be attributed to the higher content of polyunsaturated fatty acids (46.8 ± 0.2%, Appendix A) in CF compared to that in pork fat (~16%) [74]. Some insect species have been found to have antioxidant activity due to the presence of flavanols or proteins able to donate protons, terminate radicals, or chelate metal ions, but this can be lost during the pretreatment process when producing these insect ingredients, or during the cooking of the sausage itself [23,75,76,77].

#### 3.5.2. Protein Oxidation

Protein oxidation of the sausages occurred during the 14-day storage period, as indicated by the formation of carbonyls and the loss of free thiols and tryptophan FI (Figure 3b–d, Table 4). Within each sausage group, the carbonyl content showed slight increase during storage (*p* > 0.05). Contrary to carbonyls, the free thiol content decreased gradually towards the end of the storage period in each sausage group (*p* > 0.05). With the increasing CF addition, the carbonyls increased from 1.5 nmol/mg protein to 2.6 nmol/mg protein (*p* < 0.001), whereas the free thiol content significantly decreased from 127.0 ± 5.9 nmol/mg protein to 76.4 ± 2.7 nmol/mg protein on Day 0 (*p* < 0.001). However, the interaction of storage days × CF% for carbonyls and free thiols was not statistically significant (*p* > 0.05). The effect of the tryptophan FI change was significant for both storage days (*p* < 0.001) and CF addition (*p* < 0.001) (Figure 3d, Table 4). With increasing CF addition, the tryptophan FI increased significantly from 391 ± 22 to 720 ± 6 on Day 0. During the 14-day storage period, there was a dramatic decrease in tryptophan within each sausage group. Additionally, a significant interaction effect of storage day × CF% was found on the loss of tryptophan FI (*p* < 0.001).

As compared to the control sausage, there were increased levels of protein oxidation in the hybrid sausages, as evidenced by the changes in carbonyl content, free thiol levels, and tryptophan FI. This might be partly due to the increased protein content and partly due to the altered protein profiles in hybrid sausages. Muscle proteins (including actin and myosin) are susceptible to protein oxidation during storage, and actin has been identified as the predominant protein in house crickets [53]. Additionally, in hybrid sausages, there may be more exposed susceptible amino acids (lysine, arginine, histidine, tyrosine, cysteine, and methionine) from the protein amino acid side chains, which form more protein cross-links or protein carbonyls (aldehydes and ketones) upon attack by reactive oxygen species [78].

#### 3.5.3. Correlations of Oxidation Markers

A correlation between lipid and protein oxidation indicators was found for all sausages (Table 5). A high, significant, and positive correlation between carbonyls and TBARS (r = 0.566, *p* < 0.01) was observed, while free thiols showed a high, significant, and negative correlation with TBARS (r = −0.640, *p* < 0.01) and carbonyls (r = −0.854, *p* < 0.01). Most likely, this is because that the hydroperoxide (primary lipid oxidation products) can initiate protein oxidation by attacking their amino acid residues to produce, e.g., aldehydes. Vice versa, protein oxidation can also initiate lipid oxidation. Baron and Andersen (2002) [79] reviewed the myoglobin-induced lipid oxidation in muscle-based foods. It has been suggested that lipid oxidation and protein oxidation in foods interact with each other, because the free radicals and oxidative products could be transferred between lipid and protein fractions [80]. Unlike the other oxidation indicators, tryptophan seemed to have less correlation with TBARS, carbonyl, or free thiols, with the Pearson correlation coefficients being –0.243, 0.152, and –0.184, respectively. In this study, the drop in tryptophan FI occurred at a very early stage (Day 1), which was much faster than the other oxidation indicators. This is most likely because tryptophan residues are considered the most prone to rapid oxidative degradation [78]. To retard lipid and protein oxidation, both food additives (e.g., nitrite, phosphates, ascorbates) and natural extracts (from plants such as spices, herbs, berries, and fruits, either alone or as a mixture) can be used in sausages [78,81].

## 4. Conclusions

This study shows that the protein solubility and water-holding capacity of CF were significantly affected by pH and NaCl concentration, while the gelation of CF was affected by the CF content in the medium at the same pH and heating condition. The addition of CF, even in small amounts, can result in unfavorable changes to color and textural characteristics, as well as diminish the oxidative storage stability of hybrid sausages. Further research focusing on improving the texture and slowing down lipid and protein oxidation in hybrid sausages containing insect ingredients is suggested.

## Figures and Tables

**Figure 1 foods-12-01262-f001:**
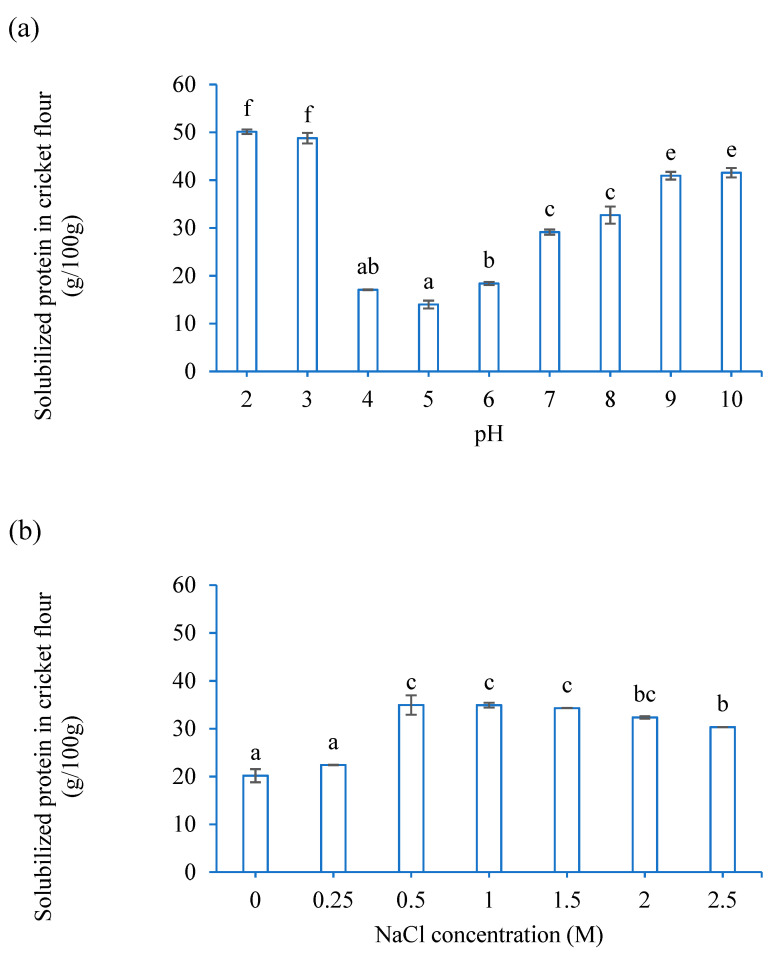
(**a**) Effect of pH on protein solubility, (**b**) effect of NaCl concentration on protein solubility, (**c**) effect of NaCl concentration on water-holding capacity. Significance differences (*p* < 0.05) are denoted by different lowercase letters a, b, c, d, e, f.

**Figure 2 foods-12-01262-f002:**
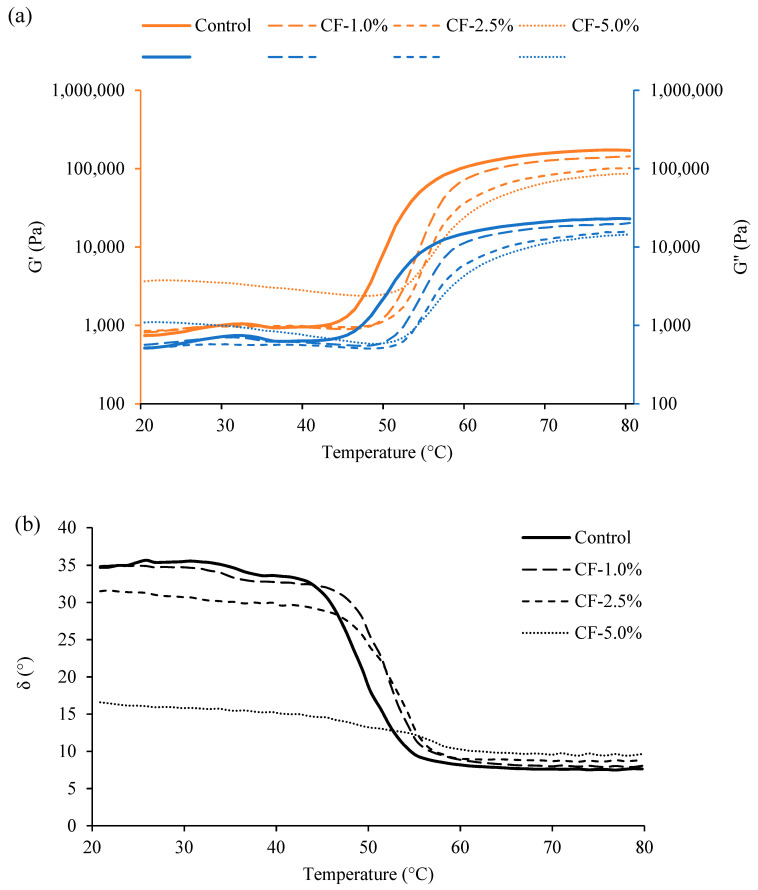
Rheological measurements (**a**) G′ (the storage modulus) and G″ (loss modulus), (**b**) δ (phase angle) of raw sausage batter formulated with lean pork, pork fat, and cricket flour when subjected to heating (20–80 °C). Control sausages are formulated with lean pork, pork fat, salt, phosphate, and ice water; CF sausages are hybrid sausages formulated by adding CF at 1% (CF-1.0%), 2.5% (CF-2.5%), and 5.0% (CF-5.0%) levels to the base (control) recipe.

**Figure 3 foods-12-01262-f003:**
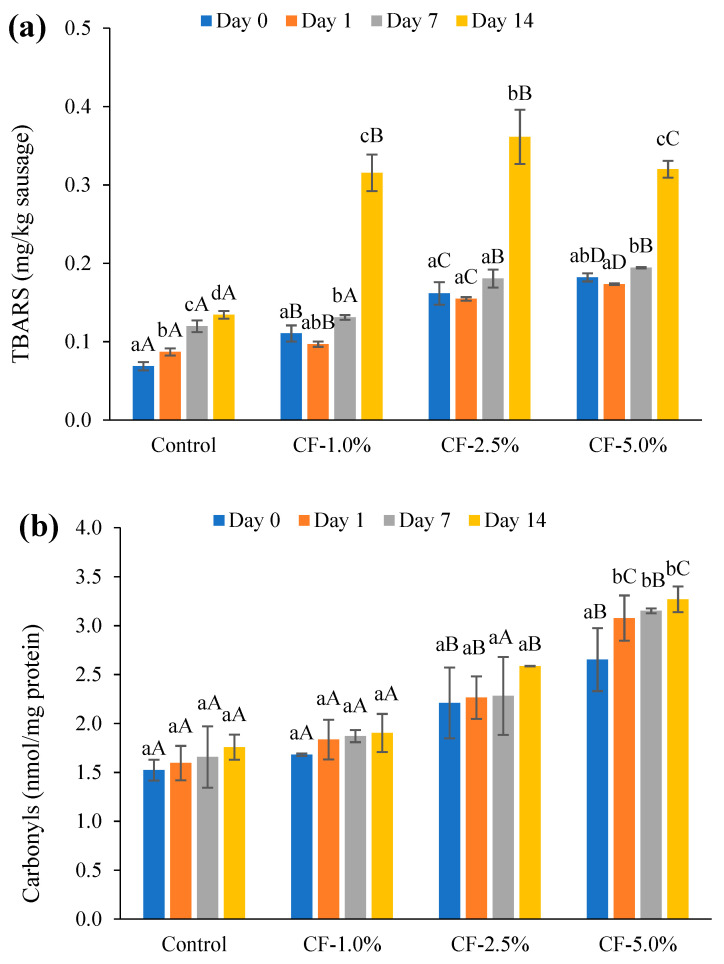
Oxidation of lipids and proteins of the cooked sausages during 14 days of storage (dark, 4 °C). (**a**) TBARS, (**b**) carbonyls, (**c**) free thiols, and (**d**) tryptophan fluorescence intensity. The control sample represents the sausage formulated with lean pork and pork fat. Control sausages are formulated with lean pork, pork fat, salt, phosphate, and ice water; CF sausages are hybrid sausages formulated by adding CF at 1% (CF-1.0%), 2.5% (CF-2.5%), and 5.0% (CF-5.0%) levels to the base (control) recipe. Significance differences (*p* < 0.05) are denoted by lowercase letters for different days within each sausage group, and by capital letters for different sausage groups on each day.

**Table 1 foods-12-01262-t001:** Ingredient amounts (g) of the formulated sausages with lean pork, pork fat, and cricket flour.

Ingredients	Control	CF-1.0%	CF-2.5%	CF-5.0%
Lean pork	2100 (35.0%)	2100 (34.7%)	2100 (34.1%)	2100 (33.3%)
Pork fat	1200 (20.0%)	1200 (19.8%)	1200 (19.5%)	1200 (19.0%)
NaCl	120 (2.0%)	120 (2.0%)	120 (2.0%)	120 (1.9%)
Phosphate *	30 (0.5%)	30 (0.5%)	30 (0.5%)	30 (0.5%)
Ice-water	2550 (42.5%)	2550 (42.1%)	2550 (41.5%)	2550 (40.5%)
CF	-	60 (1.0%)	150 (2.4%)	300 (4.8%)
Total weight	6000	6060	6150	6300

CF: cricket flour. *: Keitto-Sitonal (Mp-Maustepalvelu Ltd., Hämeenlinna, Finland; 57% P_2_O_5_). (%): calculated by the ingredient amount/total weight.

**Table 2 foods-12-01262-t002:** Measured proximate composition of the major raw ingredients (lean pork and cricket flour) and calculated proximate composition of sausages.

		Moisture	Crude Fat	Protein	Chitin	Others ^#^
		%	%	%	%	%
Ingredients	Lean pork	74.9 ± 0.2	3.3 ± 0.8	20.3 ± 0.4	-	1.5 ± 0.4
	CF *	8.2 ± 1.3	23.5 ± 0.5	56.0 ± 0.5	8.1 ± 0.8	4.2 ± 1.0
Sausages	Control	68.7 ± 0.1	21.2 ± 0.3	7.1 ± 0.1	0.0	3.0 ± 0.2
	CF-1.0%	68.1 ± 0.1	21.2 ± 0.3	7.6 ± 0.1	0.1 ± 0.0	3.0 ± 0.2
	CF-2.5%	67.2 ± 0.1	21.2 ± 0.3	8.3 ± 0.1	0.2 ± 0.0	3.0 ± 0.2
	CF-5.0%	65.8 ± 0.1	21.3 ± 0.3	9.4 ± 0.1	0.4 ± 0.0	3.1 ± 0.2

Values are given as mean ± standard deviation (*n* = 3). CF: cricket flour. *: dry matter. ^#^: = 100 − moisture, protein, crude fat, and chitin. (The others mainly including 2.5% NaCl and phosphates, according to the sausage recipe (Table 1)). The control sausage represents sausages formulated with lean pork, pork fat, salt, phosphate, and ice water; the CF sausage represents hybrid sausages formulated by adding CF at 1% (CF-1.0%), 2.5% (CF-2.5%), and 5.0% (CF-5.0%) levels to the base (control) recipe.

**Table 3 foods-12-01262-t003:** Color and texture of cooked sausages formulated with lean pork, pork fat, and cricket flour.

		Control	CF-1.0%	CF-2.5%	CF-5.0%
Color	*L**	74.8 ± 0.3 d	71.2 ± 0.1 c	69.1 ± 0.5 b	66.0 ± 0.5 a
	*a**	5.4 ± 0.0 a	5.5 ± 0.2 a	6.5 ± 0.2 b	6.7 ± 0.4 b
	*b**	11.4 ± 0.1 a	11.7 ± 0.2 a	12.3 ± 0.2 b	12.9 ± 0.1 c
	Δ*E**		4.1 ± 0.1 a	6.5 ± 0.5 b	9.7 ± 0.9 c
Texture	Hardness (g)	743.9 ± 21.1 c	573.2 ± 19.4 b	245.7 ± 8.5 a	271.0 ± 18.0 a
	Springiness (ratio)	0.9 ± 0.0 d	0.8 ± 0.0 c	0.7 ± 0.0 b	0.5 ± 0.0 a
	Cohesiveness	0.7 ± 0.0 d	0.6 ± 0.0 c	0.5 ± 0.0 b	0.4 ± 0.0 a
	Chewiness (N)	475.7 ± 23.9 c	254.5 ± 5.5 b	80.3 ± 4.5 a	56.8 ± 2.6 a
	Resilience	0.4 ± 0.0 d	0.2 ± 0.0 c	0.2 ± 0.0 b	0.1 ± 0.0 a
	Fracturability	14.9 ± 0.5 d	13.5 ± 0.8 c	10.3 ± 0.2 b	7.4 ± 0.1 a

Values are given as mean ± standard deviation (*n* = 3); CF: cricket flour; *L**: lightness; *a**: redness; *b**: yellowness; Δ*E**: total color difference between the hybrid and control sausage ($∆E^{*}=\sqrt{∆{L^{*}}^{2}+∆{a^{*}}^{2}+∆{b^{*}}^{2}}$). (Δ*E** ≤ 1.5, color difference is considered invisible for consumers; Δ*E** ≥ 6.0, color difference is considered clearly visible [70]. Control sausages are formulated with lean pork, pork fat, salt, phosphate, and ice water; CF sausages are hybrid sausages formulated by adding CF at 1% (CF-1.0%), 2.5% (CF-2.5%), and 5.0% (CF-5.0%) levels to the base (control) recipe. Significance differences (*p* < 0.05) between sausage groups are denoted by lowercase letters.

**Table 4 foods-12-01262-t004:** Effect of storage days, CF%, and their interaction on TBARS, carbonyls, free thiol group, and tryptophan.

	TBARS	Carbonyls	Free Thiols	Tryptophan
Storage days	**	**	**	**
CF%	**	**	**	**
Storage day × CF%	**	NS	NS	**

CF: cricket flour. **: *p* < 0.01. NS: not significant.

**Table 5 foods-12-01262-t005:** Pearson correlation coefficients between TBARS, carbonyls, free thiols, and tryptophan in sausages.

Correlation	TBARS	Carbonyls	Free Thiols	Tryptophan
TBARS	1			
Carbonyls	0.566 **	1		
Free thiols	−0.640 **	−0.854 **	1	
Tryptophan	−0.243	0.152	−0.184	1

**: *p* < 0.01.

## Data Availability

Data is contained within the article or Appendix A.

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
