# Peer review of "Hybrid Sausages Using Pork and Cricket Flour: Texture and Oxidative Storage Stability"

_foods, 2023, doi:10.3390/foods12061262_

Round 1

Reviewer 1 Report

The manuscript titled “Hybrid sausages using pork and cricket flour: texture and oxidative storage stability” investigates the influence of pork sausages supplementation with cricket flour on their characteristics. The topic of the article is up-to-date and interesting. However, the manuscript needs a Major Revision.

  Line 114: It should Merck.

Lines 139-141: More details on chromatographic analyses are needed.

Line 154: Please specify casing in more detail.

 Line 279: Sentence should be stylistically corrected.

 Line 312: What do you mean by notably different? The statistical analysis results are missing.

 Line 316: When replacing part of the meat different changes may occur than in the case of meat supplementation.

Line 321: Please try to explain similar fat levels despite the fact that different doses of CF were used.

Line 336 and elsewhere: Zhang not Y. Zhang

Lines 407-410: Why did you use freeze-dried flour if you knew it had worse properties than flour prepared using alternative methods?

Considering statement in line 464 what was the purpose of your research? Did you aimed to enhance nutritional value of sausage? In the introduction you mentioned environmental reasons. However, in that case you should replace meat proteins using flour. This would allow to make better conclusions. This issue needs better clarification. The reasoning presented in lines 307-308 is not convincing.

Please consider modification of title and research reasoning with consideration of the fact that you actually tested sausage supplementation with cricket flour.

Reviewer 2 Report

The manuscript is rather weak scientifically. The article needs to be revised taking into account the following points. In this way, I think that the subject will become more understandable and the results will be better conveyed to the reader.

The importance of the subject was emphasized in the introduction. However, the reasons for the analyzes were not revealed.

How many times was the empirical research repeated? In other words, how many different productions were made? How many times different productions were carried out? Is only one batch prepared? The trial plan should be clearly stated.

·      Storage factor is included for some analyses. Why was this factor not taken into account in other analyses, such as texture analysis?

     In the statistical evaluations of the 4 graphs given in Figure 3, the lettering is incorrect. Such an assessment cannot be made. Please refer to the literature on the subject. A different lettering should be made for each factor (such as A, B, C, D and a, b, c, d). In addition, according to the results given in Table 4, the interaction of storage day x CF% for carbonlys and free thiols is not statistically significant. Why plotted for a non-important interaction? Making an evaluation based on main effects is important for a healthy interpretation of the results.

The results given in Table 2 do not indicate that the experiment was repeated 3 times. Probably an experiment was conducted and sampling was done 3 times from this trial. Standard deviations also point out to this situation.

In Table 3, lettering should be done after standard deviations, for example 74,8 ± 0,3d. 

Revisions must be made in the light of the recommendations given above. 

Round 2

Reviewer 1 Report

The authors have improved the manuscript. They provided good explanations to my questions. Nevertheless, in my opinion  the manuscript still needs Minor revision.

1. When describing the aim of the study please refer to your explatantion i.e. "the aim of this particular study was to investigate the effects of added cricket flour on the textural properties and oxidative stability of hybrid sausages in order to clarify the technological properties of insect meal as ingredient in meat products".

2. Please provide reasoning for choosing freeze-dried flour in the text of the manuscript.

Author Response

  1. When describing the aim of the study please refer to your explanation, i.e. "the aim of this particular study was to investigate the effects of added cricket flour on the textural properties and oxidative stability of hybrid sausages in order to clarify the technological properties of insect meal as ingredient in meat products".

Authors’ response: Thank you for the comment, and we try to modify the aim. In the manuscript, Lines 90-93, the text has been modified to ‘Therefore, the aim of this particular study was to investigate the effects of added cricket flour on the textural properties and oxidative stability of hybrid sausages in order to clarify the technological properties of insect meal as ingredient in meat products.’

  1. Please provide reasoning for choosing freeze-dried flour in the text of the manuscript.

Authors’ response: We are sorry that we did not react enough to this the aspect, and thank you for focusing on it again.  One sentence has been added to explain the reason of freeze-drying, in the manuscript, Lines 102-105, ‘Freeze-drying was chosen to improve the mixing and comminution of the crickets, for to avoid visible brown particles on the cut surface and the overall browning of the insect ingredients, caused by heating.’

Reviewer 2 Report

The authors have made an effort to improve the manuscript, which is well reflected in much of it. However my explanation about interactions was not evaluated. The interaction of storage day x CF% for carbonlys and free thiols is not statistically significant. An interaction that is not important should not be plotted.  Factors should be discussed separately. So the averages should be compared.  For example; 

for treatments:

control: mean+SD

CF-1%: mean+SD

CF-2.5%: mean+SD

CF-5%: mean+SD

for storage:

day 0: mean+SD

day 1: mean+SD

day 7: mean+SD

day 14 : mean+SD

Author Response

Authors’ response: We are sorry that we did not respond properly to your comment earlier. Thank you for pointing this aspect again thus giving us a possibility to further improve the paper.  As stated in the comment, the interaction of storage days × CF% for carbonyls and free thiols was not statistically significant (within Table 4).  This was the conclusion after the statistical analysis of the whole data set results used to plot Figure 3, which shows the oxidation changes due to lipids (TBARS) and proteins (carbonyls, thiols, tryptophan).  In our study, particularly carbonyls and free thiols have been chosen to indicate protein oxidation of meat products according to literature, as well as many publications within our own research group. Examples are listed below:

Bao, Y., & Ertbjerg, P. (2015). Relationship between oxygen concentration, shear force and protein oxidation in modified atmosphere packaged pork. Meat Science, 110, 174-179. https://doi.org/https://doi.org/10.1016/j.meatsci.2015.07.022 ‘

Li, B., Dong, X., Puolanne, E., & Ertbjerg, P. (2022). Effect of wooden breast degree on lipid and protein oxidation and citrate synthase activity of chicken pectoralis major muscle. LWT, 154, 112884. https://doi.org/https://doi.org/10.1016/j.lwt.2021.112884

Soglia, F., Petracci, M., & Ertbjerg, P. (2016). Novel DNPH-based method for determination of protein carbonylation in muscle and meat. Food Chemistry, 197, 670-675. https://doi.org/10.1016/j.foodchem.2015.11.038

There has been little studies investigating the interaction of storage days × CF% on carbonyls or free thiols for food products especially for protein oxidation. Thus, we studied this particularly by doing statistical analysis on the whole data set. Based on the statistical results, we concluded that the interaction of storage days × CF% on carbonyls or free thiols was not statistically significant (within Table 4). In our study, the plotting of carbonyls and free thiols over storage days for all the sausage groups are used as raw data, or evidence, to show the results of Table 4, in other words, plotting of carbonyls and free thiols are compulsory to conclude Table 4. Therefore, we would suggest plotting the results for carbonyls and free thiols as shown in Figure 3.

To indicate how each single factor affect oxidation, for Figure 3, we did the statistical analysis for each single factor according to the comment in review round 1. We used capital letter ‘ABCD’ to indicate significant difference among different sausage groups (factor: CF%), we used lowercase letter ‘abcd’ to indicate significant difference among different days within the same sausage group (factor: storage days ), which was described in the manuscript Lines 689-691, caption list of Figure 3.  In such way, the two letter systems reflect each single factor.

In the manuscript, the text in Lines 456-470 has been modified into:

‘Protein oxidation of the sausages occurred during the 14-day storage as indicated by the formation of carbonyls, loss of free thiols and tryptophan FI (Figure 3b, c and d, Table 4). Within each sausage group, the carbonyl content showed slight increase during storage (P > 0.05). On the contrary to carbonyls, the free thiol content decreased gradually towards the end of storage within each sausage group (P > 0.05). With increasing CF addition, the carbonyls increased from 1.5 nmol/mg protein to 2.6 nmol/mg protein (P < 0.001), whereas the free thiol content significantly decreased from 127.0 ± 5.9 nmol/mg protein to 76.4 ± 2.7 nmol/mg protein on Day 0 (P < 0.001). However, the interaction of storage days × CF% for carbonyls and free thiols was not statistically significant (P > 0.05). The effect towards the tryptophan FI change was significant with both storage days (P < 0.001) and CF addition (P < 0.001) (Figure 3d, Table 4). With increasing CF addition, the tryptophan FI increased significantly from 391 ± 22 to 720 ± 6 on Day 0. During the 14-day storage, there was a dramatic decrease of tryptophan within each sausage group. Additionally, a significant interaction effect of storage day × CF% was found on the loss of tryptophan FI (P < 0.001).’